# Trace Amine-Associated Receptors’ Role in Immune System Functions

**DOI:** 10.3390/biomedicines12040893

**Published:** 2024-04-18

**Authors:** Vyacheslav I. Moiseenko, Vera A. Apryatina, Raul R. Gainetdinov, Sergey A. Apryatin

**Affiliations:** Institute of Translational Biomedicine, Saint Petersburg State University, 199034 Saint Petersburg, Russia

**Keywords:** trace amine-associated receptors, TAAR, trace amines, immune system, G protein-coupled receptors, lymphoid cells, myeloid cells

## Abstract

Trace amines are a separate, independent group of biogenic amines, close in structure to classical monoamine neurotransmitters such as dopamine, serotonin, and norepinephrine that include many products of the endogenous or bacteria-mediated decarboxylation of amino acids. A family of G protein-coupled trace amine-associated receptors (in humans, TAAR1, TAAR2, TAAR5, TAAR6, TAAR8, and TAAR9) that senses trace amines was discovered relatively recently. They are mostly investigated for their involvement in the olfaction of volatile amines encoding innate behaviors and their potential contribution to the pathogenesis of neuropsychiatric disorders, but the expression of the TAAR family of receptors is also observed in various populations of cells in the immune system. This review is focused on the basic information of the interaction of trace amines and their receptors with cells of the general immune systems of humans and other mammals. We also overview the available data on TAARs’ role in the function of individual populations of myeloid and lymphoid cells. With further research on the regulatory role of the trace amine system in immune functions and on uncovering the contribution of these processes to the pathogenesis of the immune response, a significant advance in the field could be expected. Furthermore, the determination of the molecular mechanisms of TAARs’ involvement in immune system regulation and the further investigation of their potential chemotactic role could bring about the development of new approaches for the treatment of disorders related to immune system dysfunctions.

## 1. Introduction

Trace amines (TAs) in mammals are generally synthesized endogenously by the decarboxylation of amino acids and the constitutive microbiota containing bacterial decarboxylases, as well as entering the body with nutritional products involving bacterial fermentation [1]. Trace amines are also found in many food products in concentrations in ranges of milligrams per kilogram, and in particular, the Mediterranean diet includes many foodstuffs containing trace amines [1,2]. The catabolism of trace amines is carried out under the action of the enzymes, monoamine oxidases A and B (MAO-A and MAO-B). In this regard, the so-called “cheese” syndrome has been described, including hypertension and headaches, that occurs after the excessive consumption of foods containing trace amines (cheese, red wine, chocolate, etc.) in patients taking MAO inhibitors [1,3]. The name “trace” reflects the relatively low content level of these substances in mammalian tissues [4]. The first and most studied of the trace amines is β-phenylethylamine (PEA) which was discovered in 1876 in the laboratory of Dr. Marceli Nencki when studying the breakdown of chicken egg proteins [5]. During this process, PEA is formed from phenylalanine under the influence of bacterial decarboxylases. This group of biogenic amines also includes p-tyramine (TYR), p-octopamine (OCT), p-synephrine (SYN), tryptamine (TRY), and several other amines.

Many biogenic amines, such as tyramine, are present in nanomolar concentrations in blood plasma and in the central nervous systems (mainly in the neurons) of healthy people [6]. The decarboxylation of amino acids (the main endogenous mechanism of TAs’ formation) occurs in the brain with great intensity and this fact suggests the participation of trace amines in the pathogenesis of neuropsychiatric disorders. Nevertheless, to determine the molecular mechanisms of the effect of nutrients of a protein nature and endogenous products of amino acid decarboxylation on the level of TAs in the brain, it is necessary to use knockout animals (for genes encoding TAARs). In clinical practice, the determination of TA levels in the blood may become diagnostically significant [1].

As of today, there are six known human functional receptors (six genes) associated with trace amines (trace amine-associated receptors, TAARs) that belong to the family of G protein-coupled receptors (GPCRs) [1]. So far, TAAR1 has been studied the most primarily for its role in the regulation of brain function via the regulation of the operation of the major monoaminergic neurotransmitter systems, such as those that are dopaminergic and serotoninergic. It is known that the other trace amine-associated receptors (TAAR2-TAAR9) are represented in the olfactory system but also that at least some of them are found in the limbic brain areas where they participate in the regulation of emotional behaviors and adult neurogenesis [7,8,9]. TAs are also represented in the peripheral organs and tissues, including the pancreas [10] and various types of immune system cells [1,11]. Furthermore, the potential role of TAARs in the development of oncological processes has been indicated [12,13,14]. As of today, there are accumulating amounts of data on the expression of the TAAR family of receptors in different populations of immune system cells; however, little is known about the regulatory role of these receptors in inflammation processes. This article overviews the available information on the potential role of trace amines and their receptors in human and other mammalian immune system cells that express the genes of TAARs.

## 2. The History of TAARs’ Discovery

The new family of GPCRs, known as trace amine system receptors, was first discovered in 2001 by two independent groups of researchers [15,16]. In 2005, a new standardized nomenclature system was proposed for TAARs [17]. 

Various species of animals are distinguished by the number of genes encoding these receptors. In zebrafish, over 100 TAARs have been identified. The flying fox has 26 identified functional TAAR genes. Bottle-nosed dolphins are the only vertebrates in whom functional TA receptor genes have not been found [18]. Nine TAAR genes have been found in humans (*TAAR1–9*), but there are six that are functionally active: *TAAR1*, *TAAR2*, *TAAR5*, *TAAR6*, *TAAR8*, and *TAAR9* [19]. *TAAR3*, *TAAR*4, and *TAAR7* are pseudogenes and thus are not functional in humans [20].

The human TAAR genes are located on the 6q23.2 chromosome [15,16]. TAAR1, among the TAAR family of receptors, has been studied the most. The receptor is expressed both in the central nervous system, where it regulates dopamine, serotonin, and glutamate neurotransmitter systems, and in the peripheral organs and tissues, including immune system cells. Currently, TAAR1 is being studied as a potential therapeutic target in the treatment of various mental disorders, such as schizophrenia [1]. 

TAAR1 is expressed in the central nervous system in the ventral tegmental area (VTA), substantia nigra, dorsal raphe nucleus (DRN), amygdaloid body, renal cortex of the medial temporal lobe, base of the hippocampus (subiculum), prefrontal cortex, nucleus accumbens, hypothalamus, cerebrospinal nucleus of the trigeminal nerve, the nucleus of the solitary tract and the medulla oblongata vomiting center [15,20,21,22,23,24]. In the peripheral organs and tissues, there is known TAAR1 expression in the β cells of the Langerhans islets, pancreas, mucous membrane of the stomach, intestine, white fatty tissues, spine, as well as in a variety of immune cells. The other functional TAAR isoforms are represented predominantly in the olfactory system where they perform a chemosensory function of sensing innate odors; however, in the brain and peripheral tissues, the expression of these receptors also appears, including in immune system cells [1,7,8,9].

Over the past few years, there has been an increase in scientific publications showing good promise for the trace amine system in biomedicine, including uses in drug development, cosmetics, dietary supplements, and specialty foods. The role of TAs in the control of behavior, energy metabolism, and cellular immune responses, including their interaction with the microbiota, in the biochemical transformations of nutrients in the body, and, as a result, in the pathogenesis of alimentary-dependent diseases, was shown [1].

Increasing evidence from preclinical and clinical studies indicates bidirectional interactions within the brain–gut–microbiome axis [25]. Microbes in the gut can communicate with the brain through at least three parallel and interacting channels involving nervous-, endocrine-, and immune-signaling mechanisms. There are several observations implicating alterations in brain–gut–microbiome communication in the pathogenesis and pathophysiology of irritable bowel syndrome, obesity, and several neuropsychiatric disorders [25]. Many TAAR ligands are produced in the gut by microbiota and the prominent expression of TAARs is found in the gut, in immune cells, as well as in the brain [1,4]. Further studies are necessary to understand how trace amines and their receptors contribute to this complicated interaction of several physiological systems. Furthermore, it would be important to increase our understanding of the potential contribution of trace amines and TAARs to the pathophysiology of gastrointestinal disorders. Intriguingly, it was found that TAAR1 may be implicated in the pathogenesis of inflammatory bowel disease [1]. Further, it has been suggested that TAAR1 may serve as a novel therapeutic drug target to be further investigated for the treatment of comorbid gut, immunological, and neuropsychiatric disorders [26].

## 3. Expression Profile of TAARs and Immune Function

The endogenous ligands activating TAAR1 are β-phenylethylamine, p-tyramine, and tryptamine, while trimethylamine (TMA), a tertiary amine and a product of the microbial degradation of carnitine and choline, is the best-known agonist of TAAR5 [27]. Dopamine metabolite 3-methoxytyramine (3-MT) and the thyroid hormone metabolite 3-iodotyronamine (T1AM) are also endogenous TAAR1 agonists [1]. T1AM has also been reported to be a TAAR5 inverse agonist [21]. Several synthetic ligands for TAAR1 and TAAR5 have also been identified [1]. At the same time, dopamine and serotonin (5-HT) show partial agonism towards TAAR1 [1,17]. Several psychotropic substances, including amphetamine-like compounds, display a high affinity for TAAR1 as well [1,28]. 

Some synthetic compounds are non-selective TAAR1 agonists, for example, imidazoline receptor ligands (clonidine, idazoxane, and guanabenz) [29], apomorphine [30], ractopamine [31], and others [1].

To date, only one selective TAAR1 antagonist has been described in detail. This is N-(3-ethoxyphenyl)-4-(1-pyrrolidinyl)-3-(trifluoromethyl)benzamide (EPPTB). This substance is more effective against mouse TAAR1 compared to that of rat and human receptors. There is the assumption that EPPTB may be an inverse agonist [32,33].

There is accumulating data on the capacity of agonists of TAARs to affect the immune system’s cell functions. For example, it has been reported that PEA, TYR, and T1AM are capable of acting as chemo-attractants, stimulating the migration of neutrophils, intensifying the secretion of IL-4 by T-lymphocytes and the production of IgE by plasmocytes [34,35,36]. Other evidence of the potential role of TAARs in immune cell functions is summarized in Table 1.

## 4. Immune Function of TAARs in Lymphoid Cells

### 4.1. B-Lymphocytes

TAAR1 and TAAR2 are predominantly expressed in B-lymphocytes, while TAAR5, TAAR6, and TAAR9 are also present there but to a lesser degree [34]. B-lymphocytes play a key role in inflammation development, in part due to their involvement in IgE synthesis. TAAR1 and its closest relative TAAR2 are also found in blood polymorphonuclear cells (PMNs) and T cells. Both receptors are co-expressed in a subpopulation of PMNs, where they are involved in the chemosensory migration toward the TAAR1 agonists PEA, tyramine, and T1AM. Furthermore, siRNA-guided experiments have shown that TAAR1 and TAAR2 are necessary for trace amine-induced blood leukocyte functions including the secretion of IgE [34]. Demonstrated TAAR1 expression in embryonic centers of B-cell maturation further supports that TAAR1 can play an important role in the immune response mediated by B-cells [37]. Interestingly, the polymorphism of the gene coding the TAAR6 receptor is linked to the effective positive action of the inhalation of corticosteroids on the treatment of bronchial asthma [42].

The study on immortalized B-cellular strains of macaque rhesus monkeys revealed a constitutively high level of TAAR1 receptor expression. This is possibly a consequence of the cellular response to the stimulating effect of the herpes virus used to create cellular strains [43,44].

Here, in the mononuclear cells of peripheral blood (PBMCs) an increase was observed in the receptor expression (with an initially low level) in response to the stimulation by the mitogen phytohemagglutinin (PHA) [43]. During methamphetamine’s stimulation of immortalized B-cells, PHA-activated lymphocytes in macaque rhesus monkeys and the HEK293 strain cells, the phosphorylation and activation of cAMP-dependent protein kinase (PKA) and protein kinase C (PKC) enzymes were observed. Increased activity of transcription factors CREB and NFAT, associated with the development of an inflammatory response, was also found [43]. The TAAR1 antagonist EPPTB had the opposite effect, inhibiting the CREB and NFAT factors. The TAAR1 expression level in B-cells may vary depending on the maturation stage and its level is higher in the circulating B-lymphocytes of blood plasma than in mature memory B-cells [12,45].

The immuno-detected protein TAAR1 was also found in normal and malignant human B-lymphocytes. The effects of TAAR1 agonists on Burkitt’s lymphoma cells of the L3055 strain were evaluated. It was observed that as a result of the effect of T1AM and o-phenyl-3-iodotyramine (o-PIT) in malignant cells, the process of apoptosis was launched as indicated by the appearance of the active form of the caspase-3 enzyme [40].

### 4.2. T-Lymphocytes

The effect of specific factors activates T-lymphocytes in different ways, stimulating their regulatory and effector functions, which determines the nature of the immune response [46]. Potula et al. [47] have shown that the effect of methamphetamine, which is a powerful agonist of TAAR1, caused oxidative stress, damage to T-cell mitochondria, and changes in their production of cytokines. Methamphetamine also increases the expression of TAAR1 mRNA in human T-lymphocytes, which leads to TAAR1-dependent Th0-to-Th2 differentiation, intensifying IL-4 production and weakening IL-2 production. At the same time, it promoted the development of inflammatory reactions as a humoral response [38]. HIV1 infection activates TAAR1 in PBMCs and this activation is intensified with amphetamine pretreatment. The activation of TAAR1 may be one of the mechanisms for the action of the virus [38]. The participation of the trace amine system could explain the manifestation of immune dysfunction in people taking amphetamine-like drugs of abuse [38,47]. 

### 4.3. NK Cells

Natural killer (NK) cells are a type of cytotoxic lymphocyte playing an important role in antiviral immunity, the recognition of malignant cells, and in the mechanisms of auto-tolerance [48]. Approximately 86.7% of the NK cells of the human leukocytic film demonstrated detectable levels of mRNA TAAR1, 2, 5, 6, and 9 that were measured using the RT-PCR technique [34]. Their specific operating mechanisms have not been investigated yet.

## 5. Immune Function of TAARs in Myeloid Cells

### 5.1. Monocytes and Macrophages

Monocytes are a group of short-lived leukocytes possessing phagocytic activity. Migrating into tissues, they are differentiated into macrophages and dendritic cells. Determining the pathogen-conservative structures with the help of pattern-recognizing receptors (such as Toll-like receptors), they are capable of phagocytizing a foreign agent and then performing an antigen-presenting function to lymphocytes. This results in an interaction between the innate and humoral immunities [49,50]. Babusyte et al. [34] have demonstrated the variance in TAAR expression in human monocytes, with 20% of the cells not expressing mRNA in any of the receptors. In the remaining cells, expression was observed in all known functional TAARs except TAAR8, and the greatest level of expression corresponds to TAAR2. 

Furthermore, in mouse bone marrow macrophages, selective TAAR1 agonist RO5256390 inhibited tumor necrosis factor (TNFα) synthesis following ATP stimulation. However, this TAAR1 agonist did not affect the ADP-induced secretion of TNFα and IL-6 in microglial cells in the mouse CNS [51].

The effect of TAAR1 ligands on mouse bone marrow-derived macrophages (BMDMs) was also studied. qRT-PCR revealed an increased expression of TAAR1 after exposure to tyramine, which is a TAAR1 agonist. The increased transcription of genes for pro-inflammatory cytokines, such as IL6, TNFα, and IL1β, was also detected. The TAAR1 antagonist EPPTB inhibited the tyramine-dependent activation of TAAR1 and inflammatory cytokine gene expression in BMDMs. Because macrophage activation is important in the pathogenesis of ulcerative colitis (UC), the authors suggested that TAAR1 may be a potential therapeutic target for UC [52].

### 5.2. Polymorphonuclear Leukocytes

Polymorphonuclear leukocytes (PMNs) are a group of immune cells that include granulocytes: neutrophils, eosinophils, and basophils. They form the first line of cellular defense because of their capacity to migrate into the inflammation’s focus using chemotaxis, and the chemo-attractants in this case are biologically active substances released by pathogens and tissue macrophages [53]. TAAR1 and TAAR2 are expressed in human polymorphonuclear leukocytes [34,37]. A chemosensory migration by the human polymorphonuclear leukocytes having these receptors, according to the trace amine concentration gradient (PEA, tyramine, and T1AM), was demonstrated. The number of migrated leukocytes from the upper into the lower system’s holes was determined using a Neubauer chamber. TAAR1 and TAAR2 are possibly not only expressed but in this case perform the function jointly, as indicated by the fact that chemotaxic migration does not occur during the neutralization of the effect of one of the genes using small interfering RNA [34]. TAAR1 expression was also found in the mast cells of mice [54] and rats [55].

### 5.3. Microglia

TAAR1 is expressed in humans in the brain dopaminergic regions, including the ventral tegmental area, substantia nigra, hippocampus, amygdala, and other major formations [15,22]. The TAAR1 receptor was also found in human astrocytes where they perform a signal function through cAMP [56]. In microglia cells, TAAR1 agonist T1AM is capable of reducing the inflammatory response stimulated by Aβ, a factor in tumor necrosis (TNFα), and by lipopolysaccharide (LPS). The inflammatory response is on the part of the microglia via the inhibition of pro-inflammatory factors’ release (IL-6, TNFα, NF-kB, MCP1 and MIP1), stimulating the release of anti-inflammatory mediators (IL-10) [39]. Interestingly, the effect of ethanol causes an increase in TAAR1 expression in human microglia cell strains HMO6, which may indicate the influence of alcohol consumption on the functioning of the immune blood–brain barrier [12,56]. The study by D’Andrea et al. [57] reported that TAAR8 transcription in astroglial cells intensifies after the effect of lipopolysaccharide. 

## 6. Immunity Pathophysiology of TAARs

Potential roles of TAARs in immunity pathophysiology are summarized in Table 2. 

A study of patients with bronchial asthma identified 15 single nucleotide polymorphisms of the TAAR6 gene [42]. Functional changes were also determined in the forced expiratory volume (FEV1) for 1 s after treatment with the inhalation of glucocorticosteroids (fluticasone). It was found that in patients who are homozygotes for the minor allele rs7772821, T > G effect of this treatment on FEV1 was considerably greater than in patients carrying the genotypes rs7772821, T/G, or T/T. These data indicate the role of TAAR6 gene polymorphism in the response of asthmatics to inhaled corticosteroid treatments [42].

TAAR ligands can be generated by the human constitutive microbiota. An association between trace amine-associated receptors and inflammatory bowel disease (IBD) has been discovered [58]. We know the role of biogenic amines in the capacity of microbiota representatives to attach to the layer of epithelial cells and penetrate it. Thus, the *Enterococcus durans* IPLA655 strain may survive in the intestinal environment and synthesize tyramine in the large intestine leading to a stronger adhesion to the intestinal epithelium and a lower Th1 activation [59]. A higher level of β-phenylethylamine in the feces could be one of the Crohn’s disease markers [60]. A role of TAAR9 in intestinal function has been demonstrated in TAAR9 knockout rats [61]. First, gene ontology enrichment analysis has revealed that in the intestine, TAAR9 is co-expressed with genes involved in intestinal mucosa homeostasis and function, including cell organization, differentiation, and death, as well as with genes implicated in dopamine signaling, which may suggest a role for this receptor in the regulation of peripheral dopaminergic transmission. Furthermore, an analysis of microbiome composition in TAAR9-KO rats revealed a significant difference in the number of observed taxa between the microbiome of TAAR9-KO rats and that in wild-type rats. In the TAAR9-KO rats, the gut’s microbial community was more variable compared to that in the wild-type rats. The research of Taquet et al. [58] detected the elevated activation of TAAR2, TAAR5, and TAAR9 genes in the material obtained from a biopsy of inflamed large intestine wall tissues in patients with Crohn’s disease. The work of Christian et al. [27] suggested a TAAR-centric hypothesis for IBD, according to which a change in the homeostasis of trace amines in the large intestine’s tissue mucous membrane could lead to immune system cell hyperactivity.

**Table 2 biomedicines-12-00893-t002:** Immunological role of the TAAR family of receptors.

Immunological Role	Receptor	Expression	Biological Function	References
Antibacterial immunity	TAAR1	-	The TAAR1 agonist tyramine intensifies the adhesion and invasion of *E. durans* in the human large intestine epithelium.	[59]
	TAAR8	Astrocytes	TAAR8 transcription in astroglial cells intensifies after the effect of lipopolysaccharide.	[57]
	TAAR1 TAAR2	Granulocytes	The effect of TAAR agonists stimulates the chemosensory migration of polymorphonuclear leukocytes.	[34]
Antiviral immunity	TAAR1	Peripheral mononuclear blood cells (PBMC).	HIV1 infection activates TAAR1 in PBMCs, the activation is intensified during the preliminary effect of amphetamine.	[38]
Bronchial asthma	TAAR6	-	The presence of single-nucleotide polymorphisms of the TAAR6 gene affects the results of treating bronchial asthma patients.	[42]
Fibromyalgia	TAAR1	-	TAAR1 gene polymorphism may be interlinked to the risk of developing fibromyalgia.	[62]
Inflammatory bowel diseases	TAAR2 TAAR5 TAAR9	Large intestine epitheliocytes	Elevated TAAR expression was found in the large intestine wall cells of patients with Crohn’s disease.	[58,60,61]

A study of over 350 genes for associations with fibromyalgia, in which 496 patients with fibromyalgia and 348 people without chronic pains (control) participated, found statistically significant differences for genes GABRB3 (rs4906902; P = 3.65 × 10^−6^), TAAR1 (rs8192619; P = 1.11 × 10^−5^), and GBP1 (rs7911; P = 1.06 × 10^−4^). The products of these genes may promote the development of this disease and be the potential target for therapy [62].

The involvement of TAAR1 in the pathophysiology of multiple sclerosis (MS) has been also investigated [63]. RT-PCR was used to study the expression of TAAR1 mRNA in CD14+ monocytes obtained from the peripheral blood of patients with multiple sclerosis. The expression level of TAAR1 in the PBMCs of patients with MS and non-inflammatory neurological diseases (NINDs) was also studied. An increase in the variance of the TAAR1 expression level was found in the PBMCs of MS and NIND patients compared to those of the control group. The authors suggested that TAAR1 expression levels may vary depending on the disease subtype. There was a significant decrease in the level of TAAR1 mRNA in the CD14+ peripheral blood monocytes of MS patients compared to those of the control group. Based on the inflammatory nature of the disease, the authors suggest the participation of TAAR1 in anti-inflammatory reactions on the part of monocytes. In vitro, in CD14+ monocyte-derived macrophages, the expression of the TAAR1 protein was found predominantly in the cell nucleus. After lipopolysaccharide (LPS) stimulation, a shift towards the diffuse intracellular localization of the receptor, presumed to be cytoplasmic, was noted. In postmortem brain sections, using immunocytochemistry and fluorescence microscopy, the TAAR1 protein was identified in the macrophages/microglia appearing in white matter and at the borders of lesions in multiple sclerosis patients. The TAAR1 staining was weaker in the lesion. The authors hypothesized that the TAAR1 protein is activated in macrophages during the active phase of extravasation and invasion into the central nervous system, which is consistent with data on the postulated role of TAAR1 in immune cells’ chemotaxis [63].

## 7. Conclusions

It is known now that TAARs are widely represented in human immune system cells. These receptors are expressed both in the cells of the lymphoid, and in the myeloid shoot of hematopoiesis. The effect of TAAR agonists models the cytokine response of T-lymphocytes, and affects the differentiation of Th-cells by regulating the type and intensity of the immune response. The joint activation of TAAR1 and TAAR2 stimulates neutrophil migration, which could indicate an important role of these receptors in the primary immune response, for example, in response to bacterial infection. TAAR agonists may stimulate IgE’s synthesis by B-cells, which could indicate their role in developing reactions of hypersensitivity and in such diseases as bronchial asthma. Intriguing data have indicated that TAARs can be involved in the chemosensory migration of immune cells towards products of the bacteria-mediated decarboxylation of amino acids.

Considering the ability of the constitutive microbiota of the human body to produce biogenic amines capable of activating TAARs, it can be hypothesized that these receptors may play a role in the development of such diseases as IBD, the pathogenesis of which has still not been fully studied. There are also publications suggesting the participation of trace amine receptors in the antitumor protection system. The capability of TAAR agonists to induce the apoptosis of Burkitt’s lymphoma cells may potentially be the basis for creating a new therapy for this disease. 

Thus, further research focused on the regulatory role of the trace amine system in the pathogenesis of immune responses and the determination of the biological mechanisms for TAARs’ actions in the immune system might bring about the development of new approaches for the treatment of diseases related to immune system dysfunctions.

## Figures and Tables

**Table 1 biomedicines-12-00893-t001:** Expression and biological function of TAARs in human immune system cells.

Receptor	Expression in Human Immune Cell Populations	Known Ligands	Biological Function	References
TAAR1	Peripheral mononuclear cells, B-lymphocytes, T-lymphocytes, polymorphonuclear neutrophils, monocyte, NK-cells	β-Phenylethylamine (PEA) 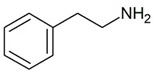	The joint effect of β-phenylethylamine and IL-4-stimulated IgE synthesis. The chemosensory migration of polymorphonuclear leukocytes towards TAAR agonists.Possible joint effect with TAAR2 due to heterodimerization.	[34]
TAAR1	Methamphetamine (METH) 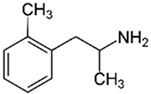	The elevated concentration of intracellular calcium, active forms of oxygen. Stimulation of the differentiation of Th0 into Th2, reduced production of IL-2, intensified production of IL-6.	[34,37,38]
TAAR1	Microglia	3-iodothyronamine (T1AM) 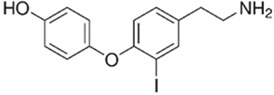	T1AM is capable of reducing the amyloid-beta (Aβ)-stimulated TNFα and LPS’s inflammatory response on part of microglia through the inhibition of the release of pro-inflammatory factors (IL-6, TNFα, NF-kB, MCP1, and MIP1), stimulating the release of anti-inflammatory mediators (IL-10)	[39]
TAAR1	Peripheral mononuclear cells, B-lymphocytes, T-lymphocytes, polymorphonuclear neutrophils, monocyte, NK-cells	Tyramine (TYR) 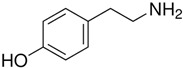 3-iodothyronamine (T1AM) 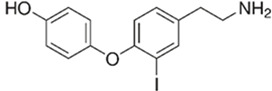	The chemosensory migration of polymorphonuclear leukocytes towards TAAR agonists.	[34,40]
TAAR2	Peripheral mononuclear cells, B-lymphocytes, T-lymphocytes, polymorphonuclear neutrophils, monocyte, NK-cells	β-Phenylethylamine (PEA) 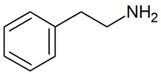	–	[34]
TAAR5	B-lymphocytes, T-lymphocytes, polymorphonuclear neutrophils, monocytes, NK-cells	Trimethylamine (TMA) 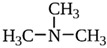 Derivative of choline 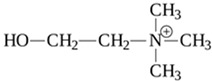	–	[11,34]
TAAR6	B-lymphocytes, T-lymphocytes, polymorphonuclear neutrophils, monocytes, NK-cells	Potent ligands have not yet been identifiedWeak activity:N-methylpiperdine 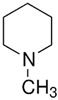	–	[36]
TAAR8	mRNA expression in leucocytes is controversial	Potent ligands have not yet been identifiedWeak activity:N-methylpiperdine 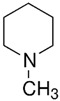 Cadaverine 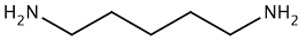	–	[34,36,41]
TAAR9	B-lymphocytes, T-lymphocytes, polymorphonuclear neutrophils, monocytes, NK-cells	Potent ligands have not yet been identifiedWeak activity:N-methylpiperdine 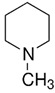 Cadaverine 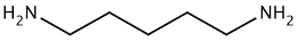	–	[36]

## Data Availability

All data listed in this review article are publicly accessible on PubMed.

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
