# Peer review of "Trace Amine-Associated Receptors’ Role in Immune System Functions"

_biomedicines, 2024, doi:10.3390/biomedicines12040893_

Round 1

Reviewer 1 Report

Comments and Suggestions for Authors

This review article is focused on summarizing current evidence for TAAR receptor expression patterns on cells of the immune system and their potential functional roles. While the concept is understudied and therefore potentially interesting to professionals of different backgrounds, the structure of the manuscript and the presentation of current evidence could be much improved. It was also noticed that the article will amply benefit by native English review, particularly for sections 4 to the end. These sections were quite hard to read.

Major recommendations are as follows:

Section 3. entitled “Ligands of the TAAR family receptors” in fact includes TAAR family receptors expression profile and potential functions in the immune system. So, the titled seems more pertinent for the second.

Perhaps expanding the information on synthetic ligands and psychotropic substances as potential regulators or tools for the future study of receptor features and function on immune cells at the end of Section 3. could make more sense.

Table 1. Second column seems unnecessary as except for TAAR1 it seems there is evidence of expression of all TAARs in the same immune cell subpopulations. Perhaps then this should be summarized in the text rather than being tabulated. However, if information about differential levels is available, I would recommend building a second table with rows for TAARs and columns for each major immune cell type, with arrows or other in the boxes where each TAAR is expressed to help readers understand differential expression of TAARs on immune subpopulations at a glance.

It is not clear why PEA is shown twice (with 2 different names same chemical structure). The horizontal lines separating METH from the second PEA and other downstream are not understood as METH and other are also TAAR1 ligands. For TAARs 6,8 and 9 the table indicates “potent” ligands not yet identified. Readers may wonder whether any weak ligands have been identified. The text “The research of D’Andrea et al. [30] revealed mRNA expression in leukocytes, however the 2013 research [24] did not confirm these results.” seems excessive for the purpose. Simply indicating “controversial” and citing both cites 24 and 30 would be enough on the table while detail explanation is provided in the text body of the article.

The link of the sentence “Interestingly, polymorphism of the gene coding the TAAR6 receptor is linked to the effective action of inhalation corticosteroids on patients with bronchial asthma [31].” To the previous sentence is not clear.

The sentence: “During methamphetamine stimulation of immortalized B-cells, PHA-activated lymphocytes in macaque rhesus monkeys, cells of the HEK293 strain,..” is confusing. It is not clear whether the evidence that is being provided was obtained from B immortalized cells, macaques or other cell types.

Please review the grammar on the following to shorten unneeded details and to clarify the message : “The effects of TAAR1 agonists on Burkitt’s lymphoma cells of the L3055 strain was evaluated. It was found that as a result of the effect of T1AM and o-phenyl-3-iodotyramine (o-PIT) in malignant cells, the process of apoptosis was launched that was validated by appearance the active form of the enzyme caspase-3 participating in this process [28].”

The sentence: “T-cells are a lymphocyte subgroup playing an important role in cellular immunity reactions” is not too informative, and the following “The effect of specific factors activates..” sentence also need to be reviewed.

Same for: “TAAR1-dependent stimulates differentiation…”

Same for: “The study in which 246 bronchial asthma patients have participated, genotyping of 15 single-nucleotide polymorphisms in the TAAR6 gene was performed. The functional changes were also determined in the forced expiratory volume (FEV1) for 1 second after treatment with inhalation glucocorticosteroids (fluticasone). It was found that in patient’s homozygote for the minor allele..”

Same for: “Data have appeared on the link be trace amine receptors..”  and “We know the role of biogenic amines in the capacity of microbiota representatives in attaching..”

Sections 4, 5 and 6 titles should indicate the focus in addition to the cell type. For example, section 4. Is not really on Lymphoid cells, it is on a particular aspect of these types of cells.

Importantly, sections 4 and following would greatly benefit from pathway diagrams that can provide the reader with an easy overview of potential or known TAAR functions in the immune vs other more studied systems. These diagrams can be easily constructed with drawing tools such as Biorender.com or similar.

Minor details:

Once an abbreviation is introduced in the main text (starting at Introduction) the extended wording should not be used. Therefore “ 2. The history of the discovery of trace amine receptors” should be  “2. The history of the discovery of TAARs”. “beta-phenylethylamine instead of its abbreviation appears on line 226. Please review the abbreviation rule throughout the manuscript.

Line 72: “With the emergence of the “brain-gut-microbiome axis,” rephrase to indicate that to clarify it refers to an emergent field or that refers to new knowledge rather than a new feature.

Line 75: “it was found that TAAR1 may be implicated..” being a narrative Review there are no new findings. There are reports of previous findings. Please, state accordingly.

Line 81: the connector “when” should be replaced by “while”.

Line 92: remove “s” from “details”

Line 115: replace “indicates further” by “further supports” or similar; and “can” by “may”.

Lines 121-122: fix line format

Line 125: remove final “e” from “mitogene”

Line 133: Is  “mature B-cells of the memory” referring mature memory B-cells?

Line 134: a space between words “also” and “found” is missing.

Line 147: remove hyphen between m and RNA

Comments on the Quality of English Language

It was also noticed that the article will amply benefit by native English review, particularly for sections 4 to the end. These sections were quite hard to read.

Author Response

REVIEWER 1

This review article is focused on summarizing current evidence for TAAR receptor expression patterns on cells of the immune system and their potential functional roles. While the concept is understudied and therefore potentially interesting to professionals of different backgrounds, the structure of the manuscript and the presentation of current evidence could be much improved. It was also noticed that the article will amply benefit by native English review, particularly for sections 4 to the end. These sections were quite hard to read.

The authors express their deep gratitude to the reviewer for his careful reading and valuable comments. We have added materials to the Introduction section and 11 more references (up to 65). The English language of the manuscript has been corrected.

Major recommendations are as follows:

Section 3. entitled “Ligands of the TAAR family receptors” in fact includes TAAR family receptors expression profile and potential functions in the immune system. So, the titled seems more pertinent for the second.

Sections 4 title has been changed (Line 121):

Expression profile of the TAARs and immune function

Perhaps expanding the information on synthetic ligands and psychotropic substances as potential regulators or tools for the future study of receptor features and function on immune cells at the end of Section 3. could make more sense.

This point has been corrected (Lines 45-53, 56-61, Table 1).

Table 1. Second column seems unnecessary as except for TAAR1 it seems there is evidence of expression of all TAARs in the same immune cell subpopulations. Perhaps then this should be summarized in the text rather than being tabulated. However, if information about differential levels is available, I would recommend building a second table with rows for TAARs and columns for each major immune cell type, with arrows or other in the boxes where each TAAR is expressed to help readers understand differential expression of TAARs on immune subpopulations at a glance.

There is not much information yet on this type of subpopulation. We would prefer to keep this table unchanged.

It is not clear why PEA is shown twice (with 2 different names same chemical structure). The horizontal lines separating METH from the second PEA and other downstream are not understood as METH and other are also TAAR1 ligands.

This point has been corrected (Table 1).

For TAARs 6, 8 and 9 the table indicates “potent” ligands not yet identified. Readers may wonder whether any weak ligands have been identified.

Thank you for your insightful suggestion. We have included weak ligands in the Table.

The text “The research of D’Andrea et al. [30] revealed mRNA expression in leukocytes, however the 2013 research [24] did not confirm these results.” seems excessive for the purpose. Simply indicating “controversial” and citing both cites 24 and 30 would be enough on the table while detail explanation is provided in the text body of the article.

It has been corrected as recommended (Table 1).

The link of the sentence “Interestingly, polymorphism of the gene coding the TAAR6 receptor is linked to the effective action of inhalation corticosteroids on patients with bronchial asthma [31].” To the previous sentence is not clear.

This point was corrected as recommended (Lines 158-160).

Please review the grammar on the following to shorten unneeded details and to clarify the message: “The effects of TAAR1 agonists on Burkitt’s lymphoma cells of the L3055 strain was evaluated. It was found that as a result of the effect of T1AM and o-phenyl-3-iodotyramine (o-PIT) in malignant cells, the process of apoptosis was launched that was validated by appearance the active form of the enzyme caspase-3 participating in this process [28].”

This point was corrected as recommended (Lines 177-180).

The sentence: “During methamphetamine stimulation of immortalized B-cells, PHA-activated lymphocytes in macaque rhesus monkeys, cells of the HEK293 strain,..” is confusing. It is not clear whether the evidence that is being provided was obtained from B immortalized cells, macaques or other cell types.

Same for: “TAAR1-dependent stimulates differentiation…”

This point was corrected as recommended (Lines 167-172 and 186-188).

The sentence: “T-cells are a lymphocyte subgroup playing an important role in cellular immunity reactions” is not too informative, and the following “The effect of specific factors activates..” sentence also need to be reviewed.

Same for: “The study in which 246 bronchial asthma patients have participated, genotyping of 15 single-nucleotide polymorphisms in the TAAR6 gene was performed. The functional changes were also determined in the forced expiratory volume (FEV1) for 1 second after treatment with inhalation glucocorticosteroids (fluticasone). It was found that in patient’s homozygote for the minor allele..”

This point was corrected as recommended (Lines 182-184 and 253-258).

The first sentence has been removed.

 The sentence: “ “The effect of specific factors activates..” we ask you to leave it unchanged.

Same for: “Data have appeared on the link be trace amine receptors…”  and “We know the role of biogenic amines in the capacity of microbiota representatives in attaching..”

This point was corrected as recommended (Line 261-263).

Sections 4, 5 and 6 titles should indicate the focus in addition to the cell type. For example, section 4. Is not really on Lymphoid cells, it is on a particular aspect of these types of cells.

Sections 4, 5 and 6 titles have been changed:

  1. Immune function of the TAARs in lymphoid cells (Line 146):
  2. Immune function of the TAARs in myeloid cells (Line 201):
  3. Immunity pathophysiology of the TAARs (Line 252):

Importantly, sections 4 and following would greatly benefit from pathway diagrams that can provide the reader with an easy overview of potential or known TAAR functions in the immune vs other more studied systems. These diagrams can be easily constructed with drawing tools such as Biorender.com or similar.

Currently, the topic of the immunological function of TAAR receptors is at an early stage of research. Considering the review and not experimental nature of the article, the authors are not ready to take responsibility for it even at the level of a hypothesis, since they consider this to be premature. It is necessary to accumulate more data and in the next review on this topic we will definitely take into account your valuable comment.

Minor details:

Once an abbreviation is introduced in the main text (starting at Introduction) the extended wording should not be used. Therefore “ 2. The history of the discovery of trace amine receptors” should be  “2. The history of the discovery of TAARs”. “beta-phenylethylamine instead of its abbreviation appears on line 226. Please review the abbreviation rule throughout the manuscript.

Line 72: “With the emergence of the “brain-gut-microbiome axis,” rephrase to indicate that to clarify it refers to an emergent field or that refers to new knowledge rather than a new feature.

Line 75: “it was found that TAAR1 may be implicated. ” being a narrative Review there are no new findings. There are reports of previous findings. Please, state accordingly.

Line 81: the connector “when” should be replaced by “while”.

Line 92: remove “s” from “details”

Line 115: replace “indicates further” by “further supports” or similar; and “can” by “may”.

Lines 121-122: fix line format

Line 125: remove final “e” from “mitogene”

Line 133: Is  “mature B-cells of the memory” referring mature memory B-cells?

Line 134: a space between words “also” and “found” is missing.

Line 147: remove hyphen between m and RNA

All minor details have been corrected.

Comments on the Quality of English Language

It was also noticed that the article will amply benefit by native English review, particularly for sections 4 to the end. These sections were quite hard to read.

The English language of the manuscript has been corrected.

Reviewer 2 Report

Comments and Suggestions for Authors

This manuscript by Vyacheslav I. Moiseenko et al., presents a “A role of trace amine-associated receptors in the immune system function” Review for TAARs expression and functional relation in the immune system. 

Here is I have minor concerns.

2-Phenylethylamine, Phenylethylamine, beta- Phenylethylamine needs to make the same as beta-Phenylethylamine. 

Inside table 1: T1AM is capable of reducing b-amyloid stimulated TNF-a  change to beta-amyloid stimulated TNF-alpha.  

Line 130: “The antagonist TAAR1” change to “The TAAR1 antagonist”.

Line 144: remove 2010 and give the reference number such as [37].

Line 144: remove 2013 and give the reference number such as [  ].

Line 172: inn to in.

Line 179: IL6, TNF-a, IL1b change to IL-6, TNF-alpha, IL-1beta.

Line 183: instead “this disease” gives more information about the disease.

Line 217: rs7772821T change to rs7772821 T (make a space).

Line 240: remove 2018 and give the reference number such as [   ].

Line 247: remove space.

Author Response

REVIEWER 2

We thank the reviewer for his valuable comments.

This manuscript by Vyacheslav I. Moiseenko et al., presents a “A role of trace amine-associated receptors in the immune system function” Review for TAARs expression and functional relation in the immune system. 

Here is I have minor concerns.

2-Phenylethylamine, Phenylethylamine, beta- Phenylethylamine needs to make the same as beta-Phenylethylamine. 

Inside table 1: T1AM is capable of reducing b-amyloid stimulated TNF-a change to beta-amyloid stimulated TNF-alpha.  

Line 130: “The antagonist TAAR1” change to “The TAAR1 antagonist”.

Line 144: remove 2010 and give the reference number such as [37].

Line 144: remove 2013 and give the reference number such as [  ].

Line 172: inn to in.

Line 179: IL6, TNF-a, IL1b change to IL-6, TNF-alpha, IL-1beta.

Line 183: instead “this disease” gives more information about the disease.

Line 217: rs7772821T change to rs7772821 T (make a space).

Line 240: remove 2018 and give the reference number such as [   ].

Line 247: remove space.

All minor details have been corrected.

Reviewer 3 Report

Comments and Suggestions for Authors

The manuscript titled: A role of trace amine-associated receptors in the immune system function was submitted by Moiseenko et al. for consideration for publication in Biomedicines. In this manuscript, the authors have reviewed the literature available on the involvement of trace amines and their associated receptors in various cells related with the immune system. The authors propose the potential use of the trace amine-associated receptors for developing novel treatments for treating diseases associated with immune dysfunction.

The manuscript adequately provides the background to the topic and the available knowledge on the role of trace amine-associated receptors in the functioning of the immune system. The most recent literature on this topic appears to be included and the authors have contextualised it.

The review is generally well presented and logical, however, the overview provided by the authors on each cell type is sometimes difficult to follow. This may be due to a language barrier. The tables summarising the function and immunological role of the various trace amine-associated receptors is a useful addition to the manuscript.

The language used by the authors indicates a lack of proficiency in the English language and detracts from the content of the manuscript. I recommend that the manuscript be edited by someone proficient in the language. In addition there are numerous inconsistencies e.g. TNF-a and TNF-α.

To date, not much has been published on trace amine-associated receptors in the field of immunology and, to this end, the authors add value to the knowledge base of this topic. In addition, the authors outline the potential use of trace amine-associated receptors as therapeutic tools for immune dysfunction. I would, following rigorous editing, recommend the manuscript for publication in Biomedicines.

Specific corrections:

1)    Page 2, Line 61 and Page 6, Line 200: ‘are’ should read ‘area’.

2)    Table 1: Correct ‘Il’ to read ‘IL’. IL is used in rest of text. Also, be consistent between TNF-a and TNF-α.

3)    Page 5, Line 125: ‘mitogene’ should read ‘mitogen’.

4)    Please look at Page 5, Lines 132 and 138. These sentences do not make sense.

5)    Page 5, Line 148: Something is missing? ‘TAAR1-dependent’ what?

6)    Page 6, Line 209: The authors mention that alcohol consumption influences the functioning of the cerebral immune barrier. Elaborate.

Comments on the Quality of English Language

The language used by the authors indicates a lack of proficiency in the English language and detracts from the content of the manuscript. I recommend that the manuscript be edited by someone proficient in the language.

Author Response

REVIEWER 3

Thank you very much for generous assessment of our manuscript.

The manuscript titled: A role of trace amine-associated receptors in the immune system function was submitted by Moiseenko et al. for consideration for publication in Biomedicines. In this manuscript, the authors have reviewed the literature available on the involvement of trace amines and their associated receptors in various cells related with the immune system. The authors propose the potential use of the trace amine-associated receptors for developing novel treatments for treating diseases associated with immune dysfunction.

The manuscript adequately provides the background to the topic and the available knowledge on the role of trace amine-associated receptors in the functioning of the immune system. The most recent literature on this topic appears to be included and the authors have contextualised it.

The review is generally well presented and logical, however, the overview provided by the authors on each cell type is sometimes difficult to follow. This may be due to a language barrier. The tables summarising the function and immunological role of the various trace amine-associated receptors is a useful addition to the manuscript.

The language used by the authors indicates a lack of proficiency in the English language and detracts from the content of the manuscript. I recommend that the manuscript be edited by someone proficient in the language. In addition there are numerous inconsistencies e.g. TNF-a and TNF-α.

To date, not much has been published on trace amine-associated receptors in the field of immunology and, to this end, the authors add value to the knowledge base of this topic. In addition, the authors outline the potential use of trace amine-associated receptors as therapeutic tools for immune dysfunction. I would, following rigorous editing, recommend the manuscript for publication in Biomedicines.

Specific corrections:

1)    Page 2, Line 61 and Page 6, Line 200: ‘are’ should read ‘area’.

2)    Table 1: Correct ‘Il’ to read ‘IL’. IL is used in rest of text. Also, be consistent between TNF-a and TNF-α.

3)    Page 5, Line 125: ‘mitogene’ should read ‘mitogen’.

4)    Please look at Page 5, Lines 132 and 138. These sentences do not make sense.

5)    Page 5, Line 148: Something is missing? ‘TAAR1-dependent’ what?

6)    Page 6, Line 209: The authors mention that alcohol consumption influences the functioning of the cerebral immune barrier. Elaborate.

 All minor details (specific corrections) have been corrected

Comments on the Quality of English Language

The language used by the authors indicates a lack of proficiency in the English language and detracts from the content of the manuscript. I recommend that the manuscript be edited by someone proficient in the language.

The English language of the manuscript has been corrected.

Reviewer 4 Report

Comments and Suggestions for Authors

Dear Authors:

The manuscript with the title „A role of trace amine-associated receptors in the immune system function” describes an very important aspects of TAARS in immune functions. Trace amines and their receptors  are an emerging pharmacological target for the treatment of human disorders and related comorbidities. While most studies have focused on their therapeutic potential for neurologic and psychiatric disorders.

This receptor family is enjoying a gradually growing interest in the therapeutic potential of this family of receptors, although TAARs were already discovered in early 2001.

For example, the currently most studied human trace amine-associated receptor 1 (hTAAR1, hTA1) is known to be a key regulator of monoaminergic neurotransmission and the actions of psychostimulants. However, expression of TAAR family receptors is observed also in various populations of cells of the immune system. The review is focused on the basic information on the interaction of trace amines and their receptors with cells of the general immune system. Only little is known about the regulatory role of these receptors in inflammation processes. The current manuscript overviews the available information on the potential role of trace amines and their receptors in the human and other mammalian immune system cells that express the genes of TAARs in a very excellent and comprehensive manner. The reader is introduced to the field by reviewing the current knowledge in a more general way. However, the provided overview is very nicely focussed and all relevant data are considered.

Due to the importance of this review article the introduction section as well as the historical overview could be supplemented by some general figures showing the different pathways or the importance of the so far considered disease areas.

Due to the emergence of the “brain-gut-microbiome axis,” the authors also take the opportunity to overview what is known about trace amines in the brain. The endogenous ligands as well as the rare identified tool compounds of the TAAR family receptors are discussed in a separate section in a quite detailed way. The tabulated overview of expression and biological function of TAARs in human immune system cells are nicely presented and summarized.

In additional sections the role of TAARs in lymphoid cells and  myeloid cells are presented. Also these sections are quite complete and comprehensive citing the most relevant literature.

Finally, the immunological role of TAAR family receptors are discussed and summarized. Also a very useful overview table is shown, which covers the current knowledge in this field.

In the final conclusion section potential involvement of TAARs in regulating the immune system is discussed. In contrast to the previous sections several speculations are provided. This section needs to be improved to differentiate more clearly between facts, which are known and validated, and the potential involvement in several diseases as a speculation, which are speculated.

Overall, I enjoyed the manuscript very much since it is a very nice summary of a less explored role of TAARs in the immune system, however, also providing an actual and comprehensive overview of the actual status. With some minor adaptions I would love to see the manuscript published in Biomedicines.

Author Response

REVIEWER  4

The authors thank the reviewer for his positive review of our review.

Dear Authors:

The manuscript with the title „A role of trace amine-associated receptors in the immune system function” describes an very important aspects of TAARS in immune functions. Trace amines and their receptors  are an emerging pharmacological target for the treatment of human disorders and related comorbidities. While most studies have focused on their therapeutic potential for neurologic and psychiatric disorders.

This receptor family is enjoying a gradually growing interest in the therapeutic potential of this family of receptors, although TAARs were already discovered in early 2001.

For example, the currently most studied human trace amine-associated receptor 1 (hTAAR1, hTA1) is known to be a key regulator of monoaminergic neurotransmission and the actions of psychostimulants. However, expression of TAAR family receptors is observed also in various populations of cells of the immune system. The review is focused on the basic information on the interaction of trace amines and their receptors with cells of the general immune system. Only little is known about the regulatory role of these receptors in inflammation processes. The current manuscript overviews the available information on the potential role of trace amines and their receptors in the human and other mammalian immune system cells that express the genes of TAARs in a very excellent and comprehensive manner. The reader is introduced to the field by reviewing the current knowledge in a more general way. However, the provided overview is very nicely focussed and all relevant data are considered.

Due to the importance of this review article the introduction section as well as the historical overview could be supplemented by some general figures showing the different pathways or the importance of the so far considered disease areas.

Due to the emergence of the “brain-gut-microbiome axis,” the authors also take the opportunity to overview what is known about trace amines in the brain. The endogenous ligands as well as the rare identified tool compounds of the TAAR family receptors are discussed in a separate section in a quite detailed way. The tabulated overview of expression and biological function of TAARs in human immune system cells are nicely presented and summarized.

In additional sections the role of TAARs in lymphoid cells and  myeloid cells are presented. Also these sections are quite complete and comprehensive citing the most relevant literature.

Finally, the immunological role of TAAR family receptors are discussed and summarized. Also a very useful overview table is shown, which covers the current knowledge in this field.

In the final conclusion section potential involvement of TAARs in regulating the immune system is discussed. In contrast to the previous sections several speculations are provided. This section needs to be improved to differentiate more clearly between facts, which are known and validated, and the potential involvement in several diseases as a speculation, which are speculated.

Overall, I enjoyed the manuscript very much since it is a very nice summary of a less explored role of TAARs in the immune system, however, also providing an actual and comprehensive overview of the actual status. With some minor adaptions I would love to see the manuscript published in Biomedicines.

Thank you very much for generous assessment of our manuscript. Unfortunately, due to limited information available at the moment we are not able to provide reasonably valid graphical images but we will certainly plan to do it in future when more data on the topic will be accumulated. All other minor details have been corrected

Round 2

Reviewer 1 Report

Comments and Suggestions for Authors

Main concerns have been correctly addressed. However, this reviewer still suggests the authors to invest the efforts to build a graphical abstract or summary pathway diagrams that can provide the reader with an easy overview of potential or known TAAR functions.

Although this is a substantially improved version, it is noticed that some typographic are still present. For example, the reviewed subtitles for subsections 4 through 6 all contain the article "the" preceding the word "TAARs" which should be removed, as the presence of this article indicates it is a subgroup of TAARs what is addressed, which is not correct. English language review is recommended before its publication.

Comments on the Quality of English Language

Although this is a substantially improved version, it is noticed that some typographic are still present. For example, the reviewed subtitles for subsections 4 through 6 all contain the article "the" preceding the word "TAARs" which should be removed, as the presence of this article indicates it is a subgroup of TAARs what is addressed, which is not correct. English language review is recommended before its publication.

Author Response

The authors sincerely thank the reviewer for his valuable comments on our article.

However, this reviewer still suggests the authors to invest the efforts to build a graphical abstract or summary pathway diagrams that can provide the reader with an easy overview of potential or known TAAR functions.

A graphical abstract has been created.

Although this is a substantially improved version, it is noticed that some typographic are still present. For example, the reviewed subtitles for subsections 4 through 6 all contain the article "the" preceding the word "TAARs" which should be removed, as the presence of this article indicates it is a subgroup of TAARs what is addressed, which is not correct. English language review is recommended before its publication.

The English language of the manuscript has been corrected

(chapter 2-6, marked in blue).